# Accurate Signal Conditioning for Pulsed-Current Synchronous Measurements

**DOI:** 10.3390/s22145360

**Published:** 2022-07-18

**Authors:** Sara Pettinato, Marco Girolami, Maria Cristina Rossi, Stefano Salvatori

**Affiliations:** 1Faculty of Engineering, Università degli Studi Niccolò Cusano, Via don Carlo Gnocchi 3, 00166 Rome, Italy; sara.pettinato@unicusano.it; 2Istituto di Struttura della Materia, Consiglio Nazionale delle Ricerche (ISM-CNR), Strada Provinciale 35D 9, Montelibretti, 00010 Rome, Italy; 3Department of Industrial, Electronic and Mechanical Engineering, Roma Tre University, Via Vito Volterra 62, 00146 Rome, Italy; mariacristina.rossi@uniroma3.it

**Keywords:** gated-integrator, synchronous measurements, radiotherapy, diamond dosimeter

## Abstract

This paper describes a compact electronic system employing a synchronous demodulation measurement method for the acquisition of pulsed-current signals. The fabricated prototype shows superior performance in terms of signal-to-noise ratio in comparison to conventional instrumentation performing free-running measurements, especially when extremely narrow pulses are concerned. It shows a reading error around 0.1% independently of the signal duty cycle (D) in the investigated D = 10^−4^–10^−3^ range. Conversely, high-precision electrometers display reading errors as high as 30% for a D = 10^−4^, which reduces to less than 1% only for D > 3 × 10^−3^. Field tests demonstrate that the developed front-end/readout electronics is particularly effective when coupled to dosimeters irradiated with the X-rays sourced by a medical linear accelerator. Therefore, it may surely be exploited for the real-time monitoring of the dosimeter output current, as required in modern radiotherapy techniques employing ultra-narrow pulses of high-energy photons or nuclear particles.

## 1. Introduction

Transducers generating an output current signal are widely employed to measure various physical quantities. Photodiodes and photoconductors, used for both photons and particle detection, are one the most important class of current transducers. Precision electrometers, typically employed to measure the output current generated by a photodetector, integrate the charge collected at the device contacts in a continuous-time mode. This approach leads to accurate measurements but requires relatively long integration periods (around hundreds of milliseconds). Therefore, when a real-time monitoring of ultra-narrow pulses is required, synchronous detection of signal is a mandatory choice. Synchronous demodulation and peak detection are widely used approaches for current signal conditioning [1,2]. Lock-in amplification is generally exploited as the most typical synchronous demodulation method allowing for high signal-to-noise ratio and good time resolution [3,4]. Gated integrator (GI), also known as “boxcar” [5,6], is a key approach to measure repetitive pulsed signals, and have already demonstrated its effectiveness in both laser beam [7] and X-ray [8] diagnostics, as well as in photoluminescence spectroscopy [9,10].

In several applications, the measurement of the integrated intensity of a current signal (e.g., the photocurrent generated by a radiation detector) is more important than the evaluation of the effective waveform. For example, in Thomson Scattering diagnostics, the information given by the integrated signal is essential to obtain the electron density and the temperature profile of plasma [11]. As a further example, modern dynamic and conformal radiotherapy techniques employ few radiation pulses releasing high dose-per-pulse values [12,13]. In this context, information on the released dose by each individual radiation pulse generated by a medical linear accelerator (LINAC), as well as the dose-rate monitoring, is crucial. In both cases, the standard continuous-integration method performed by conventional electrometers results ineffective for accurate pulse-by-pulse measurements. Moreover, the high-frequency sampling of the analog signal generated by the detector is not essential. On the other hand, the synchronous gated-integration method appears fully adequate for accurate monitoring, also in a real-time (i.e., pulse-by-pulse) diagnostics.

Lei Wu et al. [14] proposed a synchronous demodulation method based on a gated integrator. They compared the performance of the proposed system to that of lock-in amplifier for Eddy current sensor signal conditioning. Synchronous demodulation displayed only a slightly better time resolution, with a bandwidth of about 16 Hz, but with the advantages of compactness and cost effectiveness. Similarly, we also demonstrated in recent works the effectiveness of implementing synchronous integration for the dose-per-pulse monitoring of X-ray [8] and electron [15] beams employed in radiotherapy treatments.

The main advantage of GI method arises from the availability of very high precision commercially available switched integrator chips, such as the well-established IVC102, which embeds high-precision capacitors, as well as high-performance MOS switches used to reset the feedback capacitor and also allowing the hold of the output signal. These features ensure the realization of compact systems, equipped with a microcontroller for the implementation of the timing circuitry, thus largely simplifying the design. In addition to timers, the microcontroller usually integrates high-resolution successive approximation analog-to-digital converters (ADCs). Therefore, the electronics is extremely compact, and most of the design reduces to the development of a firmware tailored to the specific application.

In this work, we analyze the performance of a GI prototype for ultra-narrow current signals conditioning. Lab characterizations were performed by means of a precision current source generating pulsed signals with an extremely low duty cycle. The achieved results have been validated by field-tests carried out by acquiring the output current signals produced by a diamond dosimeter irradiated by pulsed X-rays sourced by a medical LINAC. In both cases, the results were compared to those obtained asynchronously with a commercial precision electrometer, showing better performance in terms of signal-to-noise ratio.

## 2. Materials and Methods

### 2.1. Circuit Description

Figure 1 illustrates the simplified schematic of the realized GI prototype. Integration is performed through the IVC102 chip, which demonstrates superior performance for low-level current conditioning [16,17]. For the experiments, a nominal integrating capacitance C_INT_ of 100 pF was used and obtained by the parallel connection of the three capacitors available into the chip. A 12 bits successive approximation ADC embedded into an LPC845 microcontroller with a 0–3.3 V input dynamic range was used for the analog-to-digital conversion. The front-end also includes a precision gain-selectable amplifier LT1995 as intermediate stage. Depending on the *pol* switch position, the chip is configured as either an inverting or a non-inverting amplifier. For the acquisition of positive input current signals (i.e., negative voltage ramps at the IVC102 output), the ‘a’ position is selected for the *pol* switch. Conversely, negative input currents are acquired in the ‘b’ position. In both cases, the LT1995 is configured for the unity gain. The non-inverting input of the LT1995 is biased at about 100 mV using a REF200 100 µA current reference chip. Therefore, at reset (S2 closed), the LT1995 output voltage *V*_O_ is equal to *V*_O-off_ (200 mV if *pol* = ‘a’ or 400 mV if *pol* = ‘b’). The *V*_O-off_ voltage unavoidably narrows the input dynamic range but allows for the integration of the input current pulses even in presence of relatively high offset and leakage currents, by means of the two-phase measurement method described in the following Section 2.3. For the experiments carried out in this work, only positive input currents were acquired (*pol* = ‘a’).

The state configurable timer, SCT, of the LPC845 microcontroller implements the system-timing circuitry [18,19]. The SCT generates the HOLD and RST control signals in synchronism with the SYNC input signal, adopting a reset-integrate-hold measurement method.

The LPC845 microcontroller also performs the serial communication to a computer for data and commands exchange. A specifically designed Labview^®^ virtual instrument (VI) was developed for data recording and displaying. The user-defined parameters are:the time *δt* between the rising edge of SYNC and the start of integration (both S1 and S2 closed);the integration time *T_INT_* (S1 closed and S2 open);the number *N_ACQ_* of ADC acquisitions in the hold period (both S1 and S2 open).

The three parameters are exchanged by means of the VI interface, as well as the start and stop commands of the measurement, and the download of ADC data.

### 2.2. Equipment for the Laboratory and On-Field Tests

The front-end/readout electronics used in this work was designed for the pulse-by-pulse monitoring of X-ray beams generated by a medical LINAC. For the on-field tests described in Section 3.2, a diamond dosimeter was connected to the GI input and biased at 10 V. Detector sensitivity was measured to be *S_d_* = 302.2 nC/Gy with a standard calibration procedure [20]. X-ray characterizations were performed by means of a Clinac iX (Varian Medical Systems, Inc., Palo Alto, CA, USA) installed at the Radiation Therapy Department of “San Giovanni–Addolorata” hospital, in Rome (Italy). Data acquired for pulsed output current signals generated by the detector under pulsed X-rays were recorded by means of a remotely controlled computer. The *sync* signal available at the Clinac iX’s console was used for the SYNC input of Figure 1. A detailed description of the set-up used for X-ray beams monitoring is reported in [21].

A Keithley 6221 precision current source (Keithley Instruments, Inc., Cleveland, OH, USA) was used for the lab characterizations. It was programmed to generate unipolar square-waves with very low duty-cycles (down to a few hundredths of percentage) at different amplitudes. The 6221 can generate AC signals with amplitudes in the range 2 pA–210 mA. The pulse width, Δt, can be regulated by setting the square-wave duty cycle with a 0.01% resolution. Figure 2 shows an example of the signal sourced by the 6221: a 1 mA current pulse (blue trace) with Δt = 4 µs and delayed by 12 µs from the rising edge of the active-low TTL compatible SYNC signal (green trace) provided by the 6221. The timing of the generated pulse reproduces that of the X-ray pulses emitted by the Clinac iX [18]. As highlighted by the red trace, reporting the integrated signal, the injected charge remains equal to the expected 4 nC value, despite a slight broadening of the acquired peak. Therefore, the integration performed by the proposed GI front-end in a period around the pulse duration allows for the evaluation of both the collected charge-per-pulse and the rms value of the current within the pulse period.

A Keithley 6517A electrometer (Keithley Instruments, Inc., Cleveland, OH, USA) was also used for both current and charge measurements, allowing for a comparison with the experimental results obtained with the proposed GI prototype.

### 2.3. The Implemented Two-Phase Reset-Integrate-Hold Measurement Method

Figure 3 illustrates the “two-phase” measurement method implemented by the microcontroller firmware. In this example, signals were recorded for a 4 µs wide 500 nA input pulse (i.e., injecting a charge of 2 pC) generated by the 6221 current source. The pulse delay from the rising edge of the SYNC signal was set to 12 µs. The black trace reports the LT1995 output voltage *V*_O_. The green trace is the SYNC signal. Red and blue traces are the RST and the HOLD commands, respectively, generated by the SCT and used to drive the IVC102 S1 and S2 inputs.

*(1)* 
*Pulse integration phase*


In the example reported in Figure 3, after *δ**t* = 8 µs from the SYNC rising edge, S2 becomes high, then integration starts. S1 becomes high after a further *T_INT_* = 100 µs (integration period). Then, after the end of the integration period, a hold period starts, and the ADC performs a number *N_ACQ_* of acquisitions (128 for the case reported here) defined by the user.

During the acquisitions, the microprocessor of the LPC845 accumulates the *x_i_* ADC codes in the two variables *sum* and *sum*2:(1)$$sum=\overset{N_{ACQ}}{\underset{i=1}{\sum }}x_{i}\;\;$$
(2)$$sum2=\;\overset{N_{ACQ}}{\underset{i=1}{\sum }}x_{i}^{2}\;\;$$
used to calculate both the mean value
(3)$${\bar{x}}_{1}=\frac{sum}{N_{ACQ}}\;\;$$
and the variance
(4)$$s_{1}^{2}=\frac{sum2-\frac{{\left(sum\right)}^{2}}{N_{ACQ}}}{N_{ACQ}}\;\;$$
of the of the *N_ACQ_* acquired values. During data processing, according to Equations (3) and (4), S1 and S2 are maintained at a low state. A time interval of 50 µs was set to complete data processing, also ensuring the complete discharge of the integrating capacitor.

*(2)* 
*“Zero” integration phase*


Soon after the end of data processing, the system makes a new measurement with the same timing as the first phase (i.e., the pulse integration phase): the same *δt*, *T_INT_*, and *N_ACQ_* values used during phase 1. In this second phase (called the “zero” phase), the system is able to evaluate the offset voltage around the *V*_O-off_ value, affected by both offset and leakage input currents. The microprocessor then calculates the mean value, ${\bar{x}}_{0}$, and the variance,$\;s_{0}^{2}$, of the *N_ACQ_* ADC output codes accumulated during the “zero” phase, according to Equations (1)–(4).

Both the mean value
(5)$${\bar{x}}_{p}={\bar{x}}_{1}-{\bar{x}}_{0}\;\;$$
and the variance
(6)$$s_{p}^{2}=s_{1}^{2}+s_{0}^{2}\;\;$$
related to the input pulse are finally calculated and data are sent to the computer via the serial interface.

By the observation of the black trace of Figure 3, it is worth noting that the difference between the amplitudes observed for the two phases is around the expected value of 20 mV (=500 nA × 4 µs/100 pF). Remarkably, the implemented “two-phase” method demonstrates to be effective in nulling the offset contribution recorded during the “zero” phase, which is not negligible (about 2.5 pC, i.e., the same order of the signal pulse amplitude).

### 2.4. System Calibration Procedures

Front-end calibration was performed by injecting a train of current pulses at fixed pulse repetition frequency (*PRF*) and amplitude (*I_peak_*). Injected charge packets were also changed by performing measurements at different duty cycle (*D.C*.) values. The relationship between the ADC output code, *N_ADC_*, and the *D.C.* of the input signal can be expressed as:(7)$$N_{ADC}=\frac{2^{12}}{V_{ref}C_{INT}}\left(I_{peak}\frac{D.C.}{PRF}+I_{OFF}T_{INT}\right)\;\;\;$$
where *V**_ref_* is the ADC reference voltage, *C**_INT_* the integrating capacitance, *I**_OFF_* the possible contribution of the offset current, and *T_INT_* the integration period.

The linear dependence between *D.C.* and *N_ADC_* allowed for the evaluation of the *V**_ref_*·*C**_INT_* product at fixed *I**_peak_* and *PRF* values. For the measurements, *I**_peak_* = 25 µA and *PRF* = 360 Hz were set. Figure 4a shows the mean value of the ADC codes as a function of the duty cycle from *D.C.* = 0.09% (Δt = 2.5 µs) to *D.C.* = 0.28% (Δt ≈ 7.8 µs). Each point was evaluated by averaging on 360 pulses, for a total acquisition time of 1 s. Dotted line represents the best fit linear regression line of data with a slope of 8902.9 ± 1.9, thereby estimating a *V**_ref_*·*C**_INT_* = (319.50 ± 0.07) pC. In our case, *V**_ref_* = (3.314 ± 0.001) V, as measured with a 34401A (Keysight Technologies, Santa Rosa, CA, USA) multimeter, thus evaluating *C**_INT_* = (96.42 ± 0.03) pF in good agreement with the nominal value declared for the IVC102. In addition, it is worth observing that the intercept around 0.1 indicates that the *I**_OFF_* contribution was negligible, emphasizing the effectiveness of the employed “two-phase” method.

According to the estimated *V**_ref_*·*C**_INT_* value, the VI was used to calculate the mean value and the standard deviation of the charge-per-pulse given by:(8)$$\bar{Q_{p}}=\frac{V_{ref}C_{INT}}{2^{12}}{\bar{x}}_{p}\;\;$$
(9)$$\sigma_{p}=\frac{V_{ref}C_{INT}}{2^{12}}\sqrt{s_{p}^{2}}\;\;$$
where ${\bar{x}}_{p}$ and $s_{p}^{2}$ are calculated by means of Equation (5) and Equation (6), respectively.

The rms value of the input current, *I**_rms_*, can be calculated by dividing $\bar{Q_{p}}$ of Equation (8) by the time *T* elapsed between two consecutive pulses. Therefore, the SCT was configured to capture the time *T* at each rising edge of the SYNC signal. The clock of the LPC845 microcontroller core and peripherals was set to 30 MHz, ensuring a time resolution of 33 ns. In this work, the timer-counter resolution of the SCT was set to 1 µs by dividing the clock frequency of the peripheral by 30 with the SCT prescaler. At each SYNC, the captured *T* value is inserted into the data string sent to the computer. Thus, the VI interface is able to calculate *I*_rms_ as well as the *PRF* of the train of input pulses.

To enhance the system accuracy also for time measurements, the system was calibrated by means of the 6221 current source, which ensures a frequency accuracy of ±100 ppm. Figure 4b shows a typical example of the cumulative sum of the captured SCT-counter values as a function of time. Plotted data refer to a train of 360 Hz current pulses. The best fit linear regression line of data gives a slope of 1.0065 × 10^6^ counts per second*,* corresponding to a clock frequency of about 30.2 MHz for the SCT peripheral. The rms value of the input pulse is finally calculated as:(10)$$I_{rms}=1.0065\frac{\bar{Q_{p}}}{T}\;\;$$
where $\bar{Q_{p}}$ is given by Equation (8).

## 3. Results

### 3.1. Lab Characterization

The front-end/readout electronics proposed in this work was specifically designed to be coupled to a diamond dosimeter for the real-time monitoring of pulses generated by a medical LINAC. Therefore, the 6221 current source was used to preliminary emulate in the laboratory the pulsed current signal generated by the detector under the pulsed irradiation of a Clinac iX available in the hospital. Keithley 6221 ensures a *D.C.* resolution of 0.01% and a minimum pulse duration of 1 µs [22]. The instrument was programmed to source a train of current pulses with a pulse width Δt of a few µs.

Figure 5a shows 25 µA current pulses with durations Δt = 4 µs (red trace) and Δt = 5 µs (green trace), i.e., with *PRF* = 350 Hz and *D.C*. = 0.14%, and *PRF* = 360 Hz and *D.C*. = 0.18%, respectively. The displayed *I*_rms_ values were calculated according to Equations (8) and (10) after the acquisition of more than 10^4^ pulses. The calculated mean values (34.997 nA and 45.020 nA for the 4 and 5 µs pulses, respectively) are in excellent agreement with those expected for the sourced signals. In the same figure, the acquisition performed by a Keithley 6517A electrometer is also reported (blue curve).

Figure 5b reports how the signals recorded by our GI prototype (red) and by the 6517A electrometer (blue) are spread around the expected value. Both the instruments recorded data for 2.7 h at the same input current conditions: *I*_peak_ = 25 µA, *PRF* = 350 Hz, and *D.C.* = 0.14% (Δt = 4 µs). As can be seen, the signal acquired by the synchronized GI displays a peak-to-peak amplitude of only 0.18 nA, with a 0.027 nA rms value, implying an effective resolution slightly lower than 12 bits and a noise-free resolution of 9.1 bits. Conversely, the signal recorded by the unsynchronized 6517A shows a peak-to-peak amplitude of 2.2 nA, with a rms value of 0.3 nA, i.e., more than one order of magnitude higher than those measured with our GI, thus definitely highlighting the advantages of a synchronous acquisition.

The system performance was then evaluated by injecting current pulses in the 10 nA–100 µA range. The pulse peak position was again set at about 12 µs from the SYNC rising edge and the GI was set to *δt* = 8 µs. It is worth stressing here that all the measurements were carried out with an integration time *T*_INT_ = 120 µs for both the pulse and the “zero” signals: indeed, in the nA range, the 6221 current source has a maximum settling time of 100 µs to assure an output amplitude within less than 1% of the final value [22].

Figure 6 shows the results of the calculated charge-per-pulse *Q_p_* according to Equation (8). It is worth noting the wide input dynamic range: 0.04–300 pC. Black dashed line represents the expected value of the measured *Q_p_*. The inset shows the percentage of the reading error (lower than ±1%) in the investigated dynamics, which reduced to ±0.1% in the 10–300 pC range.

To emulate the typical signal produced by a diamond detector irradiated by 6 MV pulsed X-rays [20], the 6221 was set to *PRF* = 360 Hz, *I*_peak_ = 25 µA, and *D.C.* = 0.14% (Δt ≈ 3.889 µs). In this case, the injected charge-per-pulse is about 97.22 pC. Despite the desired Δt value is out of the time resolution of the 6221, the GI clearly displays a periodic change of the pulse amplitude, as can be seen from Figure 7. For sake of clarity, only a sample of 100 pulses is reported in the figure, but the same behavior was observed for all the acquired 10^4^ pulses. In particular, the 6221 current source periodically sources 9 pulses with a calculated mean value equal to (35.003 ± 0.027) nA, in excellent agreement with the expected value of 35 nA.

Figure 8 demonstrates that a similar behavior is found when Δt is not a multiple of 0.5 µs (which is most probably the time resolution of the 6221current source).

These results highlight the pulse-by-pulse monitoring capability of the proposed GI, and provide useful insights into the peculiarity of the 6221 current source. In addition, it is worth noting that the calibration procedure described in Section 2.4 remains valid also in the presence of the periodic amplitude adjustments performed by the 6221 generator. Indeed, data reported in Figure 4a, used for the system calibration, resulted from the acquisition of 360 pulses, i.e., a multiple of the 9 pulses (corresponding to one period) observed for the 6221 current source.

### 3.2. Field-Tests

A diamond dosimeter irradiated by pulsed X-rays generated by a medical LINAC was used for the on-the-field experiments. Figure 9a reports the photogenerated current as a function of time, as recorded by means of the 6517A. The electrometer was set for high-accuracy measurements (i.e., integration time *T*_6517_ = 200 ms) and two readings per second. Measurements were performed with either 6 MV (blue trace) or 18 MV (red trace) photons at different dose rates (*DR*s). At each DR, the X-ray beam was switched on for about 60 s and then switched off for about 40 s. Acquired current amplitudes display a wide variation, more pronounced at lower *DR*s, i.e., at lower *PRF*s. As can also be observed from data of Figure 5, this effect is attributed only to an artifact due to both the impulsive nature and the very low *D.C.* of the signal (Δt = 4 µs and 30 Hz < *PRF* < 360 Hz, i.e., 1 Gy/min < *DR* < 6 Gy/min). Indeed, during a single acquisition, the instrument can record the effective current induced by either *n* − 1 or *n* + 1 pulses, where *n* = *T*_6517_ × *PRF* is the average number of pulses measured in a time interval equal to *T*_6517_. Therefore, a reading error proportional to ±1/*n* is expected. To verify this effect, Figure 9b reports the ratio between the standard deviation and the mean values of acquired data of Figure 9a as a function of *PRF* (square dots). In addition, the same figure shows the reading error as a function of *PRF* as measured by emulating the dosimeter signal by means of the Keithley 6221, sourcing Δt = 4 µs current pulses with peak amplitude *I_peak_* = 25 µA. Orange circles and green dotted line refer to reading errors on data recorded by the 6517 electrometer and the proposed prototype, respectively.

It is worth noting that the Keithley 6517 reading error is in good agreement with the expected (*T*_6517_ × *PRF*)^−1^ behavior (black dotted line). At variance, a much lower reading error is displayed by the GI prototype, resulting less than 0.11% in all the investigated *PRF* range (25–400 Hz).

Figure 10 illustrates in more details the data recorded under 6 MV pulsed X-rays with *DR* = 3 Gy/min (*PRF* = 180 Hz). The 6517A electrometer was used to perform both current and charge measurements reported in (a) and (b), respectively. Taking into account the detector sensitivity of 302.2 nC/Gy, a mean value equal to 15.11 nA for the photogenerated current is predicted for *DR* = 3 Gy/min. The values of the measured current shown in Figure 10a vary between 14.58 nA and 15.62 nA, with a mean value of 15.16 nA, in good agreement with the expected amplitude. Better results can be obtained by using the electrometer in the charge measurement mode. The collected charge as a function of time is reported as blue line in Figure 10b. Best fit of experimental data returns a slope equal to (15.120 ± 0.003) nA, in very good agreement with the predicted value. In the same figure, green dots are the values of the effective current calculated as ΔQ/ΔT, where ΔQ is the difference between two successive charge acquisitions and ΔT the acquisition time (=0.5 s). The calculated current displays a variation mainly between 14.7 nA and 15.5 nA, with a mean value of 15.12 nA. However, spikes as high as ±0.75 nA are also observed, despite the higher accuracy of the charge measurement method.

It is worth observing that the calculated current data points of Figure 10b show sharp increases followed by sharp decreases from the mean value. These peaks have almost the same amplitude with respect to the mean value indicating that integration was performed on *n* + 1 and *n* − 1 pulses, respectively. This result is further evidence that such deviations from a more stable signal amplitude are only artifacts due to the non-synchronized measurement of the current pulses generated by the detector.

Results illustrated by Figure 9 and Figure 10 underline the difficulty in making accurate current measurements of pulsed-current signals with extremely low duty-cycle.

By exploiting the LINAC’s *sync* signal, the effectiveness of the proposed GI was evaluated by connecting the prototype to the diamond dosimeter under the same irradiation conditions of measurements outlined in Figure 9 and Figure 10: 6 MV pulsed X-rays at a *DR* = 3 Gy/min (*PRF* = 180 Hz). The charge-per-pulse, *Q_p_*, reported in Figure 11 (green trace) ranges between 81.94 pC and 86.18 pC, with a mean value of 83.95 pC. Remarkably, for each pulse, ADC output codes display a variation lower than ±1.3, i.e., around ±0.1 pC for the calculated charge. Therefore, the observed 4.24 pC peak-to-peak amplitude for *Q_p_* has to be attributed to a real change of the X-ray intensity rather than to noise. This is also confirmed by the behavior of the cumulative charge, reported as a red line in the same figure. Best fit of experimental data returns a slope equal to (15.1100 ± 0.0003) nA, in excellent agreement to the expected current amplitude.

Finally, the red trace of Figure 12 shows the calculated *I*_rms_ from *Q_p_* data according to Equation (8). Current ranges between 14.75 nA and 15.51 nA, with a mean value of 15.11 nA. As observed for *Q_p_*, the ±0.38 nA variation, two times lower than what observed by 6517A current measurements and well above the ±0.018 nA system sensitivity (limited by the noise amplitude estimated during the characterizations), is attributed to the real-time change of detector irradiation conditions. To compare the amplitudes of the acquired signals, the ΔQ/ΔT values calculated by means of the 6517A electrometer in charge-mode acquisition (Figure 10b) are also shown in the same figure (blue dots) on a different timescale. As mentioned above, the ±0.75 nA peaks over the mean value are only observed for the 6517A data.

GI performance can be also compared with the results obtained with the 6517A electrometer in the same timescale. Red dots of Figure 12b are the mean values of the *I*_rms_ calculated over 90 pulses (i.e., in the same acquisition time of 0.5 s used for the 6517A). A variation lower than ±0.1 nA over the 15.11 nA mean value is detected with the GI, whereas, in the same timescale, the 6517A electrometer shows larger variation of the current amplitude (blue squares), thus highlighting the better accuracy accomplished in real-time monitoring of the proposed GI. In addition to a better time resolution, the signal recorded by means of the proposed prototype is indeed able to monitor the real intensity of the impinging radiation, without the presence of artifacts induced by the non-synchronous measurement. This feature is fundamental in X-ray beam diagnostics where the detection of irradiation anomalies is essential to the quality assurance of medical treatments. Therefore, unlike what can be observed with a conventional electrometer-based measurement technique, the proposed method is reliable, because it does not provide false over-threshold values in a pulse-by-pulse regime.

## 4. Conclusions

A synchronous pulsed current measurement method based on the gated integrator for sensor conditioning was developed and implemented in a compact prototype device. A comparative analysis between the fabricated GI prototype and a precision electrometer used for traditional current-measurement methods was provided. Characterizations were performed in a duty cycle 10^−4^–10^−3^ range for the current signals. Experimental results showed that the synchronized measurement system has undoubted advantages over the continuous integration method, as demonstrated by the very low reading error (0.1%), only attributed to the noise amplitude and significantly lower than the reading error shown by the precision electrometer operating in the same conditions (>10% for *D.C.* ≈10^−4^). This result emphasizes the drawback of unsynchronized pulsed-signal measurements: the acquisition of pulses at random times results indeed in an apparent high amplitude noise superimposed to the signal, especially for very narrow pulses.

The proposed instrument was mainly developed to provide a powerful measurement system for pulse-by-pulse measurements of current peaks generated by a diamond dosimeter under the pulsed X-rays emitted by a medical LINAC. As demonstrated by tests on-the-field, the fabricated GI prototype showed its versatility in: *(i)* measuring the charge collected at the detector contacts at each pulse; *(ii)* evaluating the instantaneous current at each peak; and *(iii)* calculating the cumulative sum of collected charge. Feature *(i)*, in particular, represents a unique property of the proposed instrument in the field of precision electrometers, by ensuring a very accurate pulse-by-pulse real-time monitoring.

The developed front-end also demonstrated excellent linearity in a wide input dynamic range of integrated charge (40 fC–300 pC). A very good sensitivity value (27 pA_rms_) was evaluated, pointing out that the equivalent number of bits is only slightly lower than the resolution of the implemented analog-to-digital converter (12 bits). In addition, the complete system was fully interfaced to a computer for both commands and data exchange.

In conclusion, the performance of the system was found to be reliable and accurate, with the advantages of compactness, low cost, and flexibility in the definition of the operation control parameters. Based on these features, the proposed instrument may surely find useful application to accurate beam diagnostics in the radiotherapy field.

## Figures and Tables

**Figure 1 sensors-22-05360-f001:**
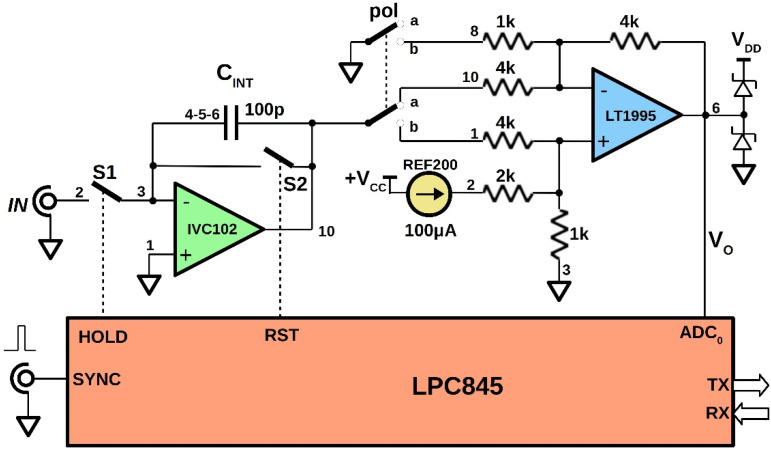
Schematic of the IVC102-based gated integrator. The LPC845 microcontroller implements the firmware used for input pulsed-current measurements synchronized with the SYNC signal.

**Figure 2 sensors-22-05360-f002:**
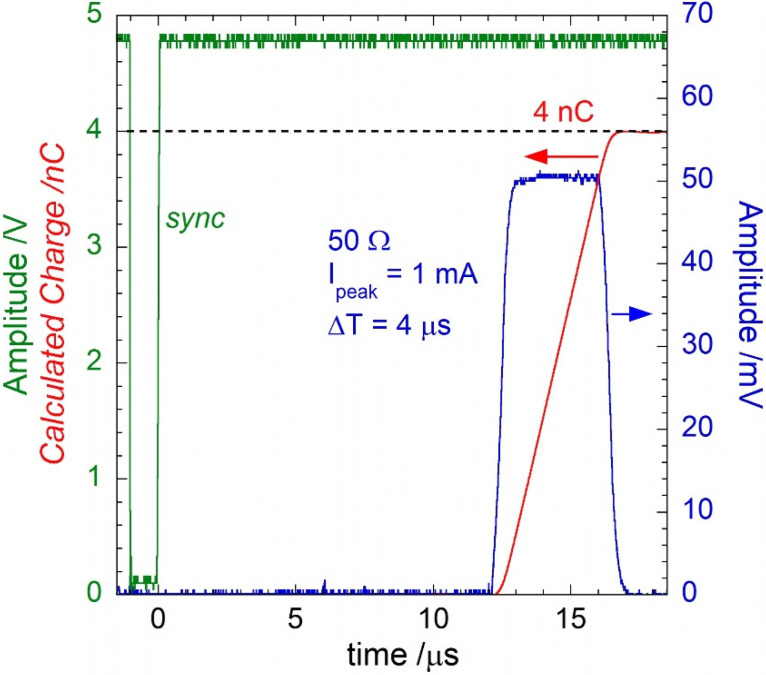
Example of a pulsed current signal (blue trace) generated by the Keithley 6221 current source. The instrument was programmed to generate a 1 mA pulse (4 µs wide) 12 µs after the sync (green trace) signal rising edge. The current signal was acquired by a 50 Ω-terminated digital oscilloscope. In red, the calculated charge over time of the current pulse.

**Figure 3 sensors-22-05360-f003:**
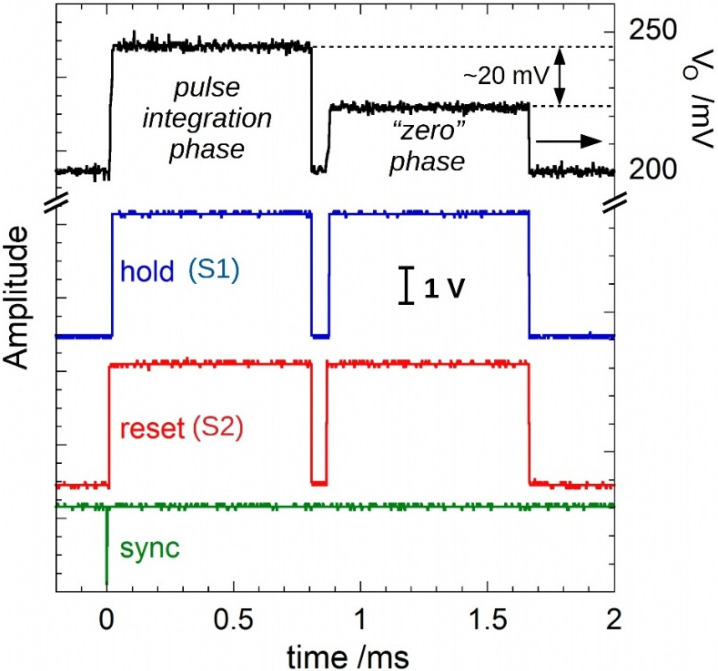
Example of signals generated for the two-phase method. The measurement during the “zero” phase allows for a strong reduction of the error induced by offset and leakage currents.

**Figure 4 sensors-22-05360-f004:**
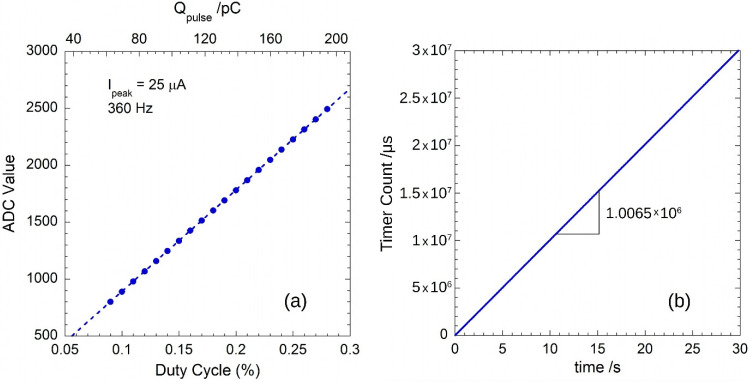
(**a**) Mean value of the ADC codes recorded for pulses at 360 Hz with a duty-cycle in the range 0.09–0.27%. The values of the injected charge pulses are shown on the top axis. Error bars are within the dimension of symbols. A *C**_INT_* = 96.42 pF is estimated by best fit (blue dotted line) of experimental data (see text); (**b**) SCT count values with a resolution of 1 µs set by means of the prescaler. Timer counts refer to the acquisitions of a train of current pulses with *PRF* = 360 Hz generated by a Keithley 6221.

**Figure 5 sensors-22-05360-f005:**
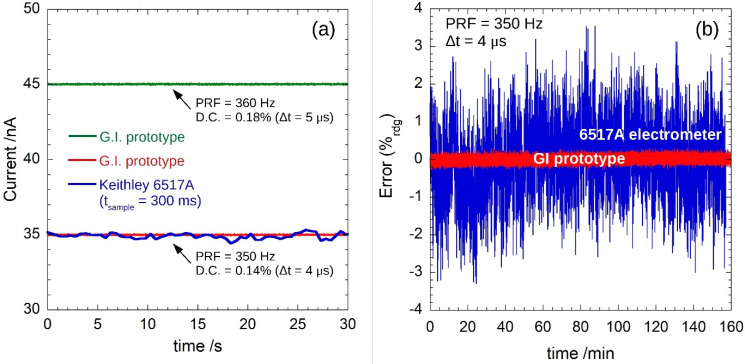
In (**a**), measurement results for an input current with *I*_peak_ = 25 µA (*PRF* and *D.C.* as reported). Red and green curves are the calculated currents according to Equations (8) and (10) of the acquisition performed by the proposed GI; in blue, the same signal recorded by a Keithley 6517A electrometer. In (**b**), the reading error on 2.7 h for the measurements performed by the GI and the 6517A electrometer for an input current of *I*_peak_ = 25 µA, *PRF* = 350 Hz, and *D.C.* = 0.14% (Δt = 4 µs).

**Figure 6 sensors-22-05360-f006:**
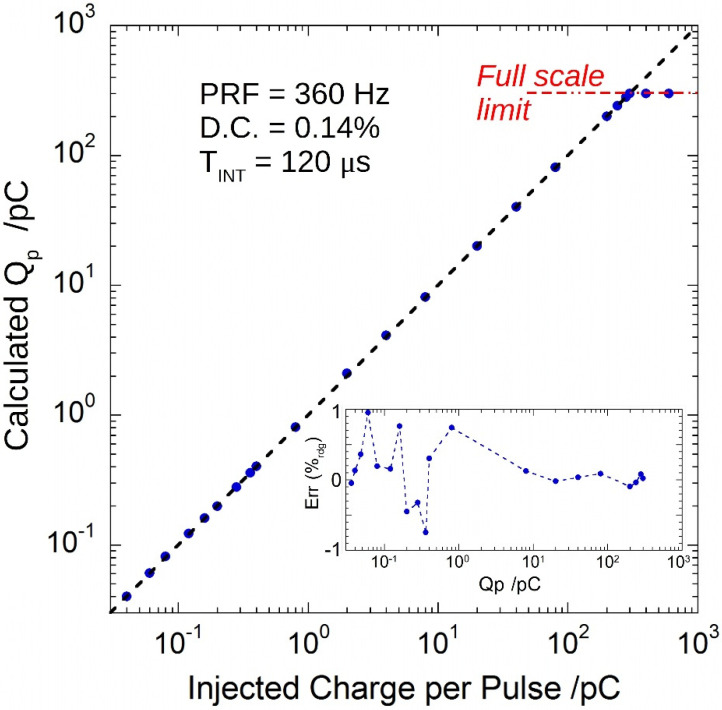
Calculated charge-per-pulse as a function of the injected pulses. Black dashed line represents the expected behavior. In the inset, the percentage of the reading error.

**Figure 7 sensors-22-05360-f007:**
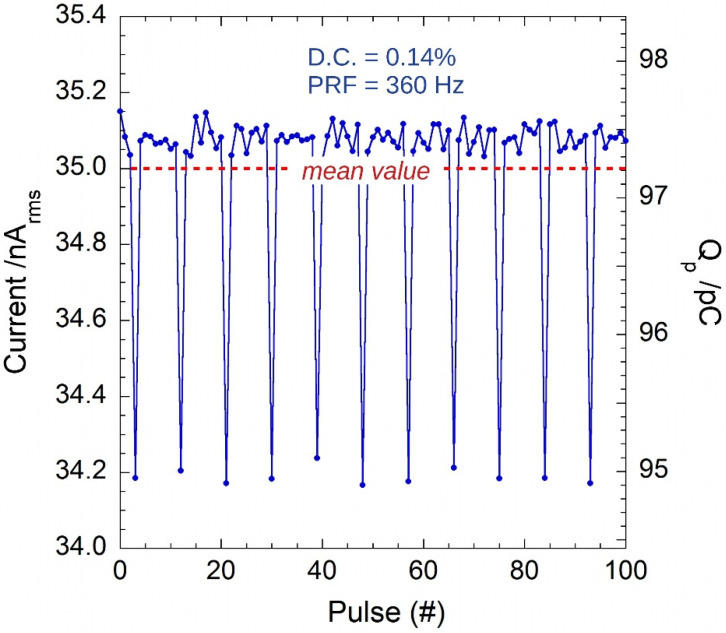
Rms current and charge-per-pulse calculated according to Equations (8) and (10) for pulses generated by the Keithley 6221 at 360 Hz. The effect of Δt not exactly equal to 4 µs is corrected by the 6221 instrument by adjusting the peak amplitudes on 9 pulses to give the programmed mean value of the effective current (red dashed line).

**Figure 8 sensors-22-05360-f008:**
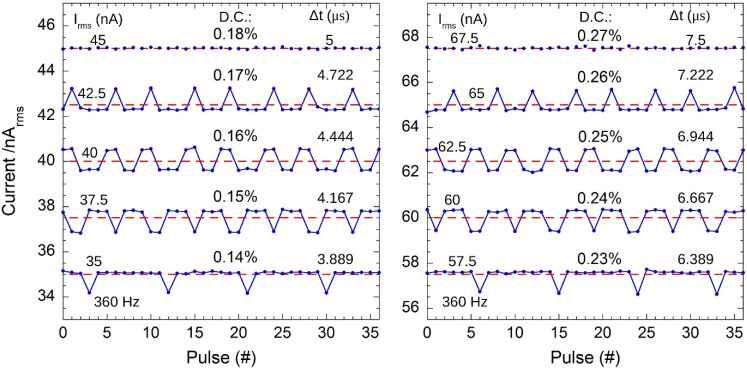
Calculated rms currents at different Δt values. The *I*_rms_ mean values are indicated as red dashed lines.

**Figure 9 sensors-22-05360-f009:**
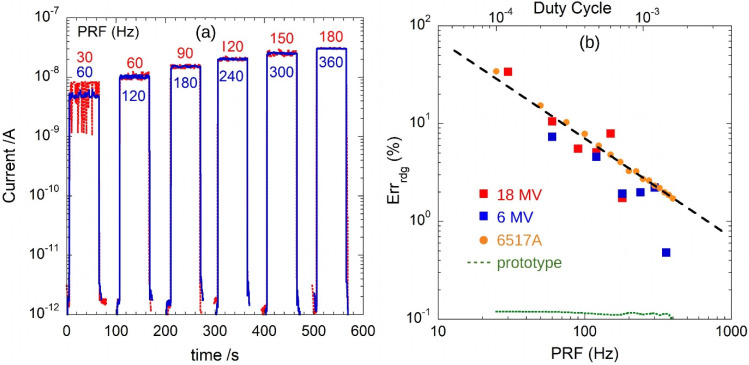
(**a**) Diamond dosimeter photocurrent under pulsed X-ray irradiations at different *PRF*, i.e., for *DR*s in the 1–6 Gy/min range. X-ray pulses are sourced by a medical LINAC at 6 MV (blue) and 18 MV (red). Experimental data were acquired by means of a Keithley 6517A electrometer. (**b**) Percentage of the reading error calculated as the ratio between the standard deviation and the mean values of data collected in (**a**) as a function of *PRF*. Orange dots (6517A electrometer) and green dotted line (proposed GI) refer to lab measurements performed with the Keithely 6221 current source (see text). Dotted line represents the expected (*T*_INT_ × *PRF*)^−1^ behavior.

**Figure 10 sensors-22-05360-f010:**
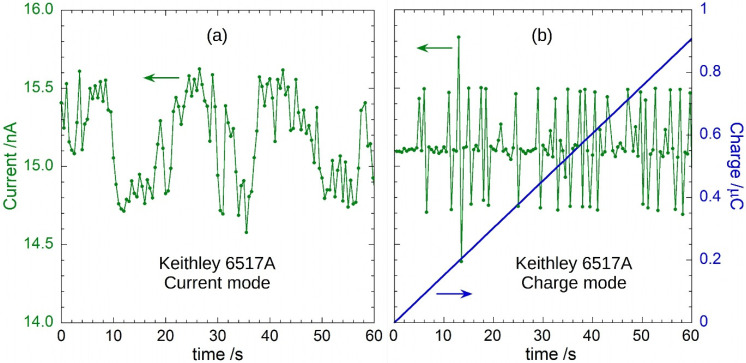
Photogenerated current and collected charge for the diamond dosimeter irradiated by 6 MV pulsed X-rays (*PRF* = 180 Hz): (**a**) current (green dots) measured by a Keithley 6517A electrometer; and (**b**) collected charge (blue line) measured by the 6517A electrometer during 1 min of diamond irradiation; green dots are calculated as ΔQ/ΔT (see text).

**Figure 11 sensors-22-05360-f011:**
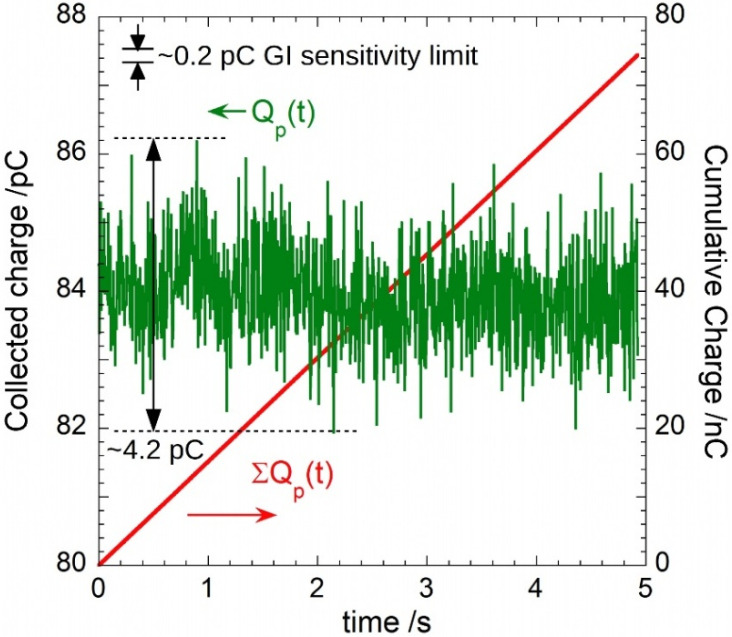
Charge-per-pulse (in green) and cumulative charge (in red) measured by the proposed GI prototype.

**Figure 12 sensors-22-05360-f012:**
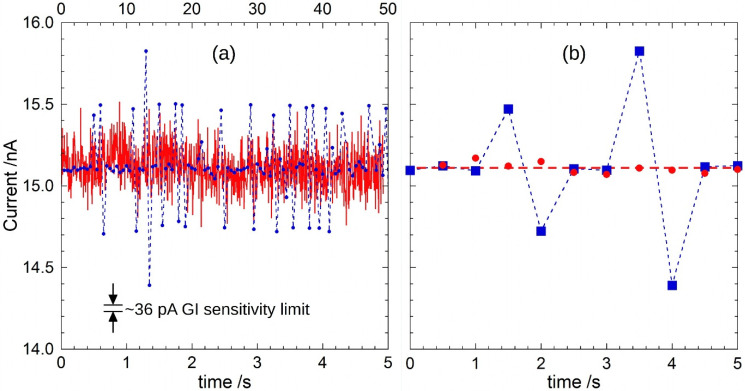
(**a**) Effective current (in red) measured by means of the GI prototype as a function of time (bottom *x*-axis). For comparison, also data of Figure 10b (blue dots) are reported on a different timescale (top *x*-axis); and (**b**) Mean values of the effective current calculated over 90 pulses, i.e., every 0.5 s (red dots). For comparison, blue squares are points acquired by 6517A electrometer in the same timescale.

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
