# Peer review of "Accurate Signal Conditioning for Pulsed-Current Synchronous Measurements"

_sensors, 2022, doi:10.3390/s22145360_

Round 1

Reviewer 1 Report

Dear Authors,

The presented results of investigation to publish in the Sensors journal are very interesting and impressive. From my point of view, the presented results can contribute to knowledge development in the area of pulsed-current sunchronous measurements.

After reading fo this manuscript, I have some comments and suggestions.

1. The manuscript is written very well. The attached images are high quality and readable.

2. When you describe the elements of the system, you should add the name of the device except its manufacturer number. For example - line 106 - LPC845 is microcontroller. It would be more understandable to the reader. Please, correct it.

3. Line 207 - The abbreviation T INT is not explained in the text.

4. Line 258 - Is "the 6221" the current source? I think that you should add the name of given value, element of the system. It will improve understanding paper.

I hope that my suggestions will help to improve the quality of this paper.

Best regards,

Reviewer

Author Response

Dear Authors,

The presented results of investigation to publish in the Sensors journal are very interesting and impressive. From my point of view, the presented results can contribute to knowledge development in the area of pulsed-current sunchronous measurements.

After reading this manuscript, I have some comments and suggestions.

1. The manuscript is written very well. The attached images are high quality and readable.

R: We are grateful to the reviewer for the comment.

2. When you describe the elements of the system, you should add the name of the device except its manufacturer number. For example - line 106 - LPC845 is microcontroller. It would be more understandable to the reader. Please, correct it.

R: Following reviewer’s suggestion, the name of the device has been added.

3. Line 207 - The abbreviation T INT is not explained in the text.

R: T_INT was defined on line 112. However, we repeated its definition for more clarity.

4. Line 258 - Is "the 6221" the current source? I think that you should add the name of given value, element of the system. It will improve understanding paper.

R: We thank the reviewer for the comment. Yes, the 6221 is the current generator. We specified it.

Reviewer 2 Report

The proposal is interesting, there are some recommendations to achieve a clearer presentation.

1.- Equation (3) and (4) are confusing, it presents the calculation of the difference of two measured values (or difference of two variances) where one is the input signal, then we would be talking about the error measurement? Explain clearly what is being calculated in this part.

2.- Equation 4, 5 and 6, indicate the minimum development to obtain these expressions, since they are fundamental in the support of the final results.

3.- What does NADC represent,

4.- It is not clear the comparative of Figure 4. It is recommended to adjust axes for a better comparison.

5.- The results of Fig 5a, should be improved to check the comparison with the Keithley 6517A electrometer, equation 5 indicates Qp=standard deviation of the charge-per-pulse, but in Fig. 5a a current value is indicated, and it is not indicated that the Keithley instrument is measuring.

Author Response

The proposal is interesting, there are some recommendations to achieve a clearer presentation.

We are grateful to the reviewer for the comment.

1.- Equation (3) and (4) are confusing, it presents the calculation of the difference of two measured values (or difference of two variances) where one is the input signal, then we would be talking about the error measurement? Explain clearly what is being calculated in this part.

R: Thanks for the comment. For eq. 3, the equations of the mean value and the variance were reported in two rows for more clarity, as well as eqs. 1 and 2. Also the text was changed accordingly. In addition, two sub-sections have been created to better clarify the adopted two-phase method. The mean value and the variance calculated during the two phases have been named with subscript ‘1’ (first phase) and subscript ‘0’ (second phase, named “zero” phase).

2.- Equation 4, 5 and 6, indicate the minimum development to obtain these expressions, since they are fundamental in the support of the final results.

R: We agree with the reviewer. We split eq. 5 into two equations: Qp for the mean value (5.1), and sigma for the standard deviation (5.2). Conversely, eq. 4, referring to the calibration procedure performed to find the Vref*CINT product, was left unchanged.

3.- What does NADC represent,

R: N_ADC is defined just before eq. 4.

4.- It is not clear the comparative of Figure 4. It is recommended to adjust axes for a better comparison.

R: Probably, reviewer refers to fig. 5. Following the comment n. 5, we added the legend for better comparison of data of fig. 5.

5.- The results of Fig 5a, should be improved to check the comparison with the Keithley 6517A electrometer, equation 5 indicates Qp=standard deviation of the charge-per-pulse, but in Fig. 5a a current value is indicated, and it is not indicated that the Keithley instrument is measuring.

R: Thank the reviewer for the observation. As indicated in the text, data of fig. 5a are calculated according to eq. 6. We added the legend specifying the measurements acquired by the GI prototype and that performed by means of the Keithley 6517.

Round 2

Reviewer 2 Report

The new proposal is suitable for publication.